# Definitive Chemoradiation Associated with Improved Survival Outcomes in Patients with Synchronous Oligometastatic Esophageal Cancer

**DOI:** 10.3390/cancers15092523

**Published:** 2023-04-28

**Authors:** Thomas Matoska, Anjishnu Banerjee, Aditya Shreenivas, Lauren Jurkowski, Monica E. Shukla, Elizabeth M. Gore, Paul Linsky, Mario Gasparri, Ben George, Candice Johnstone, David Johnstone, Lindsay L. Puckett

**Affiliations:** 1Department of Radiation Oncology, Medical College of Wisconsin, 8701 Watertown Plank Road, Milwaukee, WI 53226, USA; tmatoska@mcw.edu (T.M.);; 2Department of Biostatistics, Medical College of Wisconsin, 8701 Watertown Plank Road, Milwaukee, WI 53226, USA; 3Department of Hematology and Oncology, Medical College of Wisconsin, 8701 Watertown Plank Road, Milwaukee, WI 53226, USA; 4Department of Hospital Medicine, Washington University School of Medicine in St. Louis, 4523 Clayton Ave, CB 8058-59-01, St. Louis, MO 63110, USA; 5Department of Cardiothoracic Surgery, Medical College of Wisconsin, 8701 Watertown Plank Road, Milwaukee, WI 53226, USA

**Keywords:** oligometastatic, esophageal cancer, chemoradiation, definitive treatment

## Abstract

**Simple Summary:**

Metastatic esophageal cancer is generally treated with palliative intent, prioritizing symptom control over more aggressive treatment. Existing guidelines for metastatic esophageal cancer do not distinguish between many metastases or few: oligometastatic disease. Some research suggests that patients with oligometastatic disease, typically defined as ≤5 sites of metastatic spread, may live longer with more aggressive locoregional therapy. The purpose of this study is to report the clinical outcome of oligometastatic esophageal cancer patients treated with definitive intent chemoradiation to the primary tumor and regional nodal disease at a single institution.

**Abstract:**

Background: The study of oligometastatic esophageal cancer (EC) is relatively new. Preliminary data suggests that more aggressive treatment regimens in select patients may improve survival rates in oligometastatic EC. However, the consensus recommends palliative treatment. We hypothesized that oligometastatic esophageal cancer patients treated with a definitive approach (chemoradiotherapy [CRT]) would have improved overall survival (OS) compared to those treated with a purely palliative intent and historical controls. Methods: Patients diagnosed with synchronous oligometastatic (any histology, ≤5 metastatic foci) esophageal cancer treated in a single academic hospital were retrospectively analyzed and divided into definitive and palliative treatment groups. Definitive CRT was defined as radiation therapy to the primary site with ≥40 Gy and ≥2 cycles of chemotherapy. Results: Of 78 Stage IVB (AJCC 8th ed.) patients, 36 met the pre-specified oligometastatic definition. Of these, 19 received definitive CRT, and 17 received palliative treatment. With a median follow-up of 16.5 months (Range: 2.3–95.0 months), median OS for definitive CRT and palliative groups were 90.2 and 8.1 months (*p* < 0.01), translating into 5-year OS of 50.5% (95%CI: 32.0–79.8%) vs. 7.5% (95%CI: 1.7–48.9%), respectively. Conclusions: Oligometastatic EC patients treated with definitive CRT benefited from that approach with survival rates (50.5%) that vastly exceeded historical standards of 5% at 5 years for metastatic EC. Oligometastatic EC patients treated with definitive CRT had significantly improved OS compared to those treated with palliative-only intent within our cohort. Notably, definitively treated patients were generally younger and with better performance status versus those palliatively treated. Further prospective evaluation of definitive CRT for oligometastatic EC is warranted.

## 1. Introduction

In 2021, there was an estimated 19,260 new cases of esophageal cancer in the United States. The 5-year relative survival has trended upwards over the past 50 years to between 20–25%, though patients with metastatic disease at diagnosis have a poor prognosis, with a 5-year survival rate of 5% [1,2]. Current NCCN guidelines for metastatic esophageal cancer recommend palliative systemic therapy or best supportive care [3].

In recent years, importance has been placed on distinguishing limited metastatic disease, or oligometastatic disease, from extensive metastatic disease. With improvements in multimodality therapy, those with oligometastatic disease of various primary sites have been shown to benefit from more aggressive or definitive treatment [4,5,6,7,8]. In esophageal cancer, early reports suggest more aggressive treatment options appear to improve survival for oligometastatic patients [9,10,11,12,13]. There are varying definitions of esophageal oligometastatic disease; for the purposes of this study, it was defined as 5 or less foci of disease [9,14,15]. Nevertheless, guidelines do not currently distinguish between oligometastatic and extensive metastatic disease in esophageal cancer.

This study assessed American Joint Committee on Cancer (AJCC) 8th ed. stage IVB patients and whether the use of definitive chemoradiation in oligometastatic (≤5 sites of metastasis) esophageal cancer patients improved overall survival outcomes compared to oligometastatic patients treated palliatively within our institution [16]. We hypothesized that patients diagnosed with synchronous oligometastatic esophageal cancer treated with definitive chemoradiation to the primary site (tumor and regional lymph nodes) would have improved survival. We collected data on prognostic factors (e.g., Eastern Cooperative Oncology Group [ECOG] performance status, age) to assess which oligometastatic patients may benefit from definitive treatment.

## 2. Materials and Methods

This retrospective study was conducted from May 2021 to December 2022. The database included patients diagnosed with esophageal cancer between August 2009 and February 2020. For the purposes of this study, synchronous oligometastatic disease was pre-defined as Stage IVB having ≤5 total foci of metastasis at diagnosis. All patient information was extracted from the electronic medical records and MOSAIQ^®^ radiation oncology information system. This study was approved by an Institutional Review Board Committee prior to the start of data collection. Patients were selected for the oligometastatic cohort using the following inclusion criteria: ≥18 years old, diagnosed with AJCC 8th ed. stage IVB esophageal cancer with ≤5 sites of metastasis, and complete data regarding the initial treatment of their esophageal cancer. Exclusion criteria included no treatment received and inadequate information present in electronic medical records. Stage IVB non-oligometastatic patients were included as a comparator in some analyses.

Following patient selection, oligometastatic patients were divided into a definitive treatment group and a palliative treatment group. Definitive chemoradiation treatment was defined as ≥40 Gy to the primary tumor and surrounding lymph nodes and two cycles of chemotherapy +/− induction chemotherapy. Patients may have received subsequent radiation treatment to the metastases or subsequent doses of chemotherapy after primary definitive chemoradiation treatment. Palliative treatment was defined as those who received chemotherapy alone, radiation therapy alone regardless of radiation dose, chemoradiation with <40 Gy to the primary site or <2 cycles of chemotherapy, or another non-definitive treatment. Overall survival was calculated for each patient from diagnosis to date of death or last follow-up. Progression-free survival was also calculated for each patient from date of diagnosis to date of progression of disease or date of recurrence. Data about prognostic factors collected for each patient included age, sex, histology (adenocarcinoma vs. squamous), use of induction chemotherapy, site of the primary tumor, ECOG performance status at or near diagnosis, tumor grade, T-stage, and type of metastatic spread (hematogenous, lymphatic, direct invasion).

Descriptive statistics, including medians and ranges for continuous variables and numbers/proportions for all categorical variables, were reported. Survival functions (overall survival and progression free survival) were computed using the Kaplan–Meier method [17]. Comparison of survival curves between categorical variables were performed using the log-rank test, while survival curves between continuous variables of interest were compared using the Cox proportional hazards model [18]. All analyses considered a type I error level of 5% and wherever relevant, for multiplicity corrections, Bonferroni corrections were used to maintain this type I error level. R version 4.1.2 was used for the analysis [19].

## 3. Results

### 3.1. Patient Characteristics

Among the 78 stage IVB patients, there were 36 patients with oligometastatic disease. Of these, 27 patients (75%) had adenocarcinoma and 9 patients (25%) had squamous cell carcinoma. When patients were divided into treatment groups, 19 (52.8%) patients met criteria for definitive chemoradiation and 17 (47.2%) patients met criteria for the palliative treatment group. Of the 17 patients in the palliative treatment group, 4 patients received chemotherapy alone and 11 patients received chemotherapy and radiation <40 Gy (Range: 8 Gy–37.5 Gy) to either the primary tumor or palliative radiation to sites of metastases. Two patients received higher doses of radiation (60 Gy and 70 Gy) without any chemotherapy, which did not meet our pre-specified definition of definitive treatment. These two patients had invasion of the primary tumor into the thyroid gland. Of note, one of these patients (60 Gy) lived 67.9 months and the other patient (70 Gy) lived 3.1 months. Patient characteristics from both oligometastatic treatment groups and the 42 widely metastatic patients can be seen in Table 1.

### 3.2. Survival Analysis

The median follow-up for all oligometastatic esophageal cancer patients (*n* = 36) was 16.5 months (R: 2.3–95.0 months). The median overall survival for the definitive chemoradiation group and palliative treatment group was 90.2 and 8.1 months, respectively (*p* < 0.01) (Figure 1). The 5-year overall survival rate was 50.5% (95%CI: 32.0–79.8%) for the definitive treatment group and 7.5% (95%CI: 1.7–48.9%) for those who received palliative treatment. The median progression-free survival was 19.7 months for the definitive chemoradiation group and 5.8 months for the palliative group (*p* < 0.001) (Figure 2).

### 3.3. Comparison to Widely Metastatic Group

There were 42 patients with metastatic esophageal cancer that had ≥5 metastatic lesions and did not meet oligometastatic criteria. Demographic information for these patients can also be seen in Table 1. Four of the widely metastatic patients received definitive treatment, and the rest of the patients received palliative treatment. Within the very small cohort of widely metastatic patients who received definitive treatment (*n* = 4), no patient was still alive at 5 years while the median overall survival and progression-free survival was 14.7 and 8.6 months, respectively, using Kaplan–Meier survival estimates.

### 3.4. Prognostic Factor Analysis

The results of the univariate analyses comparing prognostic factors to overall survival are summarized in Figure 3. Factors that were associated with improved overall survival in our oligometastatic esophageal cancer cohort on univariate analysis included ECOG performance status of 2 and 3, lower age (continuous variable), adenocarcinoma histology (vs. squamous cell carcinoma), and the use of induction chemotherapy. Factors that were not significantly associated with overall survival in oligometastatic esophageal cancer included T stage, tumor grade at biopsy, type of spread (hematogenous vs. lymphatic vs. local invasion), sex, and location of primary tumor.

## 4. Discussion

Metastatic esophageal cancer has long been associated with poor survival and palliative regimens remain the standard of care [3]. Nevertheless, growing recent data suggest that select oligometastatic esophageal cancer patients may have a survival benefit with more aggressive treatment [9,10,11]. Within our cohort, synchronous oligometastatic esophageal cancer patients treated with locoregional definitive chemoradiation had significantly higher overall survival compared to those treated palliatively. A limitation in comparing these two groups is those treated more aggressively inherently had better performance status and younger age. Those attributes very likely contribute to the differences seen in survival. However, it also shows that for those appropriately selected for definitive treatment, 5-year overall survival reached 50.5%, representing a stark difference from the 5% rate reported nationally for all (unstratified) metastatic patients [1].

The study of oligometastatic cancers is still relatively new, with limited data available regarding esophageal cancer, specifically. However, some studies investigating different treatment regimens for oligometastatic esophageal cancer have been published in recent years. Chen et al. retrospectively analyzed a cohort of stage IVB patients with predominantly squamous cell carcinoma and ≤3 sites of metastases. The definitive chemoradiation group (50 Gy to primary tumor and 45 Gy to all metastases) had a statistically significant higher progression free survival than the chemotherapy alone group (8.7 months vs. 7.3 months, respectively) (*p* = 0.002), and overall survival was numerically higher in the definitive chemoradiation group (16.8 months vs. 14.8 months, respectively); however, this result did not reach significance (*p* = 0.056) [10]. Additionally, a recent retrospective study was published by Shi et al. investigating definitive dose concurrent chemoradiotherapy (50 Gy to primary tumor and 45 Gy to metastatic sites) in squamous cell carcinoma histology oligometastatic (≤5 metastatic sites) esophageal cancer patients. The chemoradiotherapy group (*n* = 240) had significantly improved overall survival (median 18.5 months) and progression free survival (median 9.7 months) compared to chemotherapy alone. Both Chen et al. and Shi et al. may have lower overall survival and progression free survival than our study due to differences in patient selection, histology, and treatment paradigms. Further study into selection criteria for the use of definitive treatment is necessary.

The efficacy of palliative treatment regimens, such as chemotherapy alone, radiation therapy alone, and chemotherapy + radiation therapy, has all been investigated in metastatic esophageal cancer, including widely metastatic [20,21,22]. Early data has also supported that more aggressive approaches in certain metastatic esophageal cancer patients may improve survival. Guttmann et al. data showed definitive chemoradiation (defined as >50.4 Gy + chemotherapy) increased survival of metastatic (including widely metastatic) esophageal cancer patients compared to chemotherapy alone and chemotherapy + palliative radiation (11.3 months vs. 8.3 months vs. 7.5 months, respectively) [23]. Within our small cohort of patients who received definitive dose treatment for widely metastatic disease (*n* = 4), no patients were alive at 5 years. Median overall survival and progression-free survival for these few was 14.7 and 8.6 months, respectively, using Kaplan–Meier survival estimates. As these estimates represent only a few patients, caution should be applied to any comparisons. These results affirm the need to investigate more aggressive treatment options in a range of metastatic esophageal cancer patients.

Lower ECOG performance status was associated with improved overall survival in oligometastatic patients within our cohort (*p* < 0.001). The prior literature has also shown that metrics such as lower ECOG performance status, lower Charlson comorbidity score, and lower WHO performance score are associated with better overall survival in oligometastatic and metastatic esophageal cancer [11,23,24]. Other studies have found no correlation between these metrics and overall survival [10]. Other significant predictors of overall survival on univariate analysis in our oligometastatic esophageal cancer cohort included younger age, receiving induction chemotherapy prior to radiation, and adenocarcinoma histology (vs. squamous cell carcinoma). The significant association between these factors and overall survival suggests further prospective study is required to further explore these factors’ effects on prognosis.

We observed an unusually favorable survival among those treated definitively. This may, in part, be explained by an on-average younger and healthier patient population at a tertiary cancer center. Our experience has allowed us to consider this approach for those who are heathier than average metastatic esophageal patients. Historically, those who are younger or of better performance status have not been treated aggressively. Even the definition of oligometastatic disease varies immensely nationally and internationally. Additionally, differences in what constitutes metastatic lesions or lymph nodes, as well as treatment protocols, may also vary among institutions, causing a difference in overall survival.

Adenocarcinoma of the esophagus is more prevalent than squamous cell carcinoma in the United States [3]. In the study cohort, the oligometastatic patients had predominately adenocarcinoma (75%) and higher overall survival compared to the squamous cell carcinoma. Interestingly, historically in locally advanced esophageal cancer, squamous cell carcinoma patients had longer survival compared to adenocarcinoma [25]. Many of our adenocarcinoma patients received induction chemotherapy in their definitive treatment. The role of induction in oligometastatic treatment remains an interesting question to be explored. Others have suggested a benefit in squamous cell populations as well [11], but this was not seen in our small cohort. A larger sample and randomized data are needed to explore this question.

Patients within this study were largely treated with intensity-modulated radiation therapy (IMRT). External beam radiation techniques and technology have improved over the last several decades, moving away from conventional two-dimensional radiation techniques. Image-guided radiation therapy has also been essential in allowing for safer treatment with less toxicity, especially in cases where there may be larger than average treatment volumes to treat disease. Within our study we did not dictate treatment to areas other than the primary and regional nodes. In cases where the oligometastasis was adjacent (e.g., non-regional abdominal node, solitary spinal metastasis), it was sometimes included in primary volumes. Given the diversity in approaches, and that some patients had radiation to distant oligometastases synchronously, immediately sequentially, long after treatment to the primary, or not at all, we did not do any sub-analyses in this area. Treating all sites of disease remains an interesting area under investigation. Improved radiation techniques have undoubtedly led to these explorations through more precise delivery of radiation to the target volume(s) in more advanced disease.

Limitations of our study include the retrospective nature and sample size. Study size limited our analysis to univariate analysis; multivariate analysis was not conducted due to small sample sizes. The definitively treated oligometastatic patients had, on average, younger patients with more favorable performance status and these variables were unable to be controlled for. Thus, it is difficult to distinguish if definitive vs. palliative treatment affects survival over performance status and age in our oligometastatic population. The large difference in survival between oligometastatic patients treated definitively vs. palliatively generates the hypothesis that there is a role for definitive chemoradiation in select oligometastatic esophageal cancers, and further study is required to validate this approach. Another limitation is that our study is a single-institution study and results may not be generalizable to other centers that have different patterns of care or care for a different subpopulation of esophageal cancer patients. Further multi-institutional studies are warranted. One final limitation of our study was the inclusion of two patients who received definitive dose radiation therapy in our palliative treatment group, as they did not receive chemotherapy, as previously discussed. Consideration of high-dose radiation could be included in future studies, but our numbers were too small to adequately assess.

The AIO-FLOT trial and EA2183 trial are prospective, randomized, phase-3 trials evaluating more aggressive treatment regimens in oligometastatic esophageal cancer. The AIO-FLOT (RENAISSANCE) trial aims to evaluate patients with oligometastatic gastric and gastroesophageal junction cancer to receive 4 cycles of chemotherapy alone (with trastuzumab if Her2+). Afterwards, those without disease progression are randomized to additional chemotherapy or surgical resection of the primary and metastases, followed by subsequent chemotherapy [26]. The EA2183 trial also begins with 4 cycles of chemotherapy, and those who do not progress are randomized to receive radiation to all sites of disease or systemic treatment alone [27]. Several other trials are also underway that include oligometastatic esophageal cancer [28,29,30].

## 5. Conclusions

In this study, oligometastatic esophageal cancer patients treated with definitive chemoradiation to the primary disease and regional lymph nodes had improved overall survival and progression free survival compared to oligometastatic patients treated with palliative intent. While the oligometastatic patients who received definitive chemoradiation were younger and had better performance status compared to oligometastatic patients receiving palliative treatment, the patients receiving definitive treatment had 5-year overall survival outcomes (50.5%) that were higher than the national 5-year overall survival (5%) for people with metastatic esophageal cancer that did not delineate between oligometastatic or widely metastatic disease. Hence, these results suggest further investigation into definitive chemoradiation treatment for appropriately selected oligometastatic esophageal cancer patients.

## Figures and Tables

**Figure 1 cancers-15-02523-f001:**
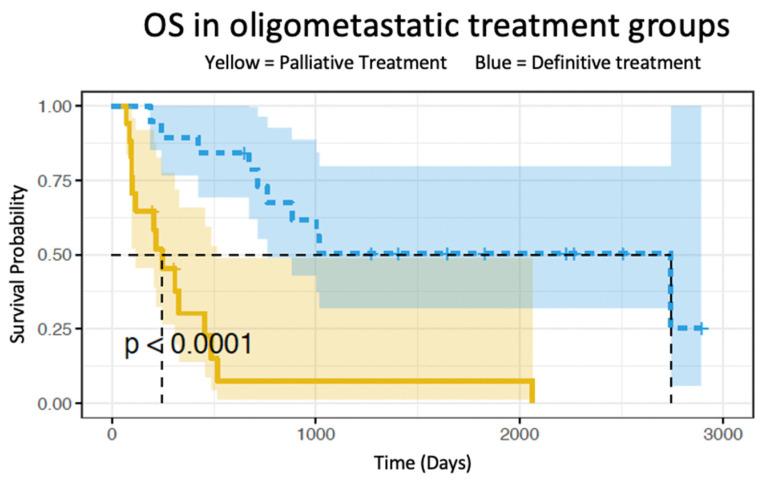
Survival graph comparing oligometastatic overall survival in patients receiving definitive chemoradiation (blue) and oligometastatic patients receiving palliative treatment (yellow).

**Figure 2 cancers-15-02523-f002:**
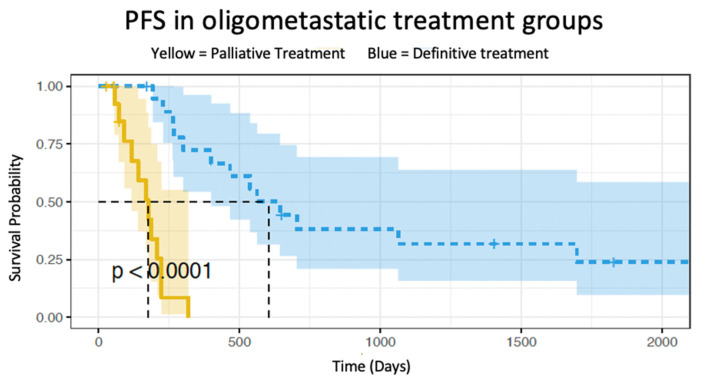
Survival graph comparing oligometastatic progression free survival in patients receiving definitive chemoradiation (blue) and oligometastatic patients receiving palliative treatment (yellow).

**Figure 3 cancers-15-02523-f003:**
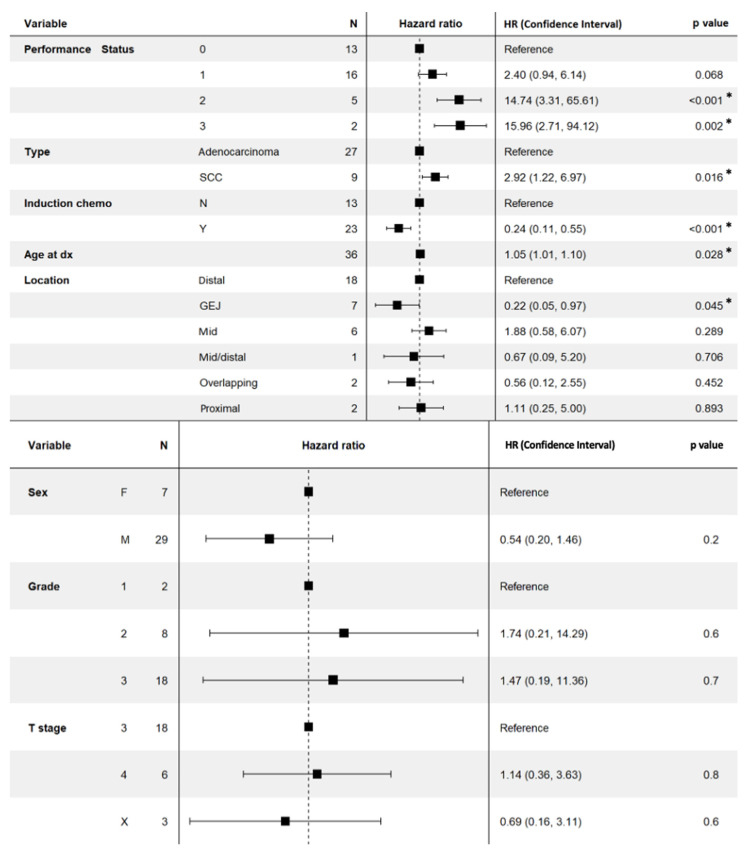
Prognostic factors association with overall survival based on univariate analysis. A “*” sign indicates *p*-value < 0.05 indicating the prognostic variables significant association with overall survival on univariate analysis. Hazard ratios were included in parentheses for prognostic factors that were significantly associated with overall survival. Abbreviations: Dx = diagnosis, HR = Hazard ratio, SCC = squamous cell carcinoma, GEJ = gastroesophageal junction, M = male, F = female, * = statistically significant.

**Table 1 cancers-15-02523-t001:** Demographic information for all treatment groups observed in this study.

Treatment Group	Oligometastatic Definitive CRT (*n* = 19)	Oligometastatic Palliative Tx (*n* = 17)	Widely Metastatic (Any Tx, *n* = 42)
Median age at diagnosis (Years)	65	69	60
Age range	30–81	52–93	26–81
Male	18 (94.7%)	6 (35.3%)	37 (88.1%)
Female	1 (5.3%)	11 (64.7%)	5 (11.9%)
Adenocarcinoma	17 (89.5%)	10 (58.8%)	35 (83.3%)
Squamous Cell Carcinoma	2 (10.5%)	7 (41.2%)	7 (16.7%)
Diagnosed prior to 2013	5 (26.3%)	7 (41.2%)	10 (23.8%)
Diagnosed in 2013 or after	14 (73.7%)	10 (58.8%)	32 (76.2%)
Average ECOG Status (0–4)	0.42 (0–1)	1.41 (0–3)	1.04 (0–3)
Median OS (months)	90.2	8.1	11.7

Abbreviations: CRT = chemoradiation; Tx = treatment; OS = overall survival.

## Data Availability

Data available on request due to privacy restrictions. The data presented in this study are available on request from the corresponding author. The data are not publicly available due to the database containing patient identifiers.

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
