# Peer review of "Definitive Chemoradiation Associated with Improved Survival Outcomes in Patients with Synchronous Oligometastatic Esophageal Cancer"

_cancers, 2023, doi:10.3390/cancers15092523_

Round 1
Reviewer 1 Report
This study investigates the efficacy of definitive chemoradiotherapy (CRT) for patients with oligometastatic esophageal cancer (EC). The researchers hypothesized that aggressive treatment may improve survival in select patients, despite the consensus recommendations for palliative treatment. The study retrospectively analyzed patients treated in a single academic hospital and divided them into definitive and palliative treatment groups. Definitive CRT was defined as radiation therapy to the primary site with >40 Gy and >2 cycles of chemotherapy.
Of the 78 Stage IVB patients, 36 met the pre-specified oligometastatic definition, and 19 received definitive CRT, while 17 received palliative treatment. The study found that patients treated with definitive CRT had significantly improved overall survival (OS) compared to those treated with palliative intent and historical controls. The median OS for the definitive CRT group was 90.2 months, and for the palliative group, it was 8.1 months (p<0.01). The 5-year OS for the definitive CRT group was 50.5%, while for the palliative group, it was only 7.5%.
The study concludes that oligometastatic EC patients treated with definitive CRT benefited from the approach, with survival that vastly exceeded historical standards for metastatic EC. The results suggest that definitive CRT should be considered for select patients with oligometastatic EC, rather than exclusively palliative treatment. However, the study's limitations include its retrospective nature and the small sample size of patients. Moreover, the patients in the definitive CRT group were generally younger and had better performance status, which could have influenced the results.
Overall, this study provides valuable insights into the treatment of oligometastatic EC and suggests that more aggressive treatment may be beneficial in select patients. However, further studies with larger sample sizes and more rigorous designs are needed to confirm these findings.
In addition, i have several comments to improve the manuscript.
1. i am confused by the definition of dCRT. Definitive chemoradiation treatment was defined as (≥ 40 Gy to the primary tumor and surrounding lymph nodes and two cycles of chemotherapy +/- induction chemotherapy). What is the treatment for the metastases?
Palliative treatment was defined as thosewho received chemotherapy alone, radiation therapy alone.
High dose Radiation therapy can be considered curative in selective cases.
The Overall survival of patients with stage IV esophageal cancer is very suprisingly and much higher than previously reported. please comment on these findings.
Table 2 prognostic factors are the associated with overall survival, please provide the HR, now it unclear if it associated with good or bad prognosis.
Reviewer 2 Report
This study aimed to determine whether treating patients with oligometastatic esophageal cancer (EC) using a definitive approach (chemoradiotherapy) would improve overall survival compared to purely palliative treatment. The study analyzed 36 patients with oligometastatic EC, 19 of whom received definitive treatment and 17 received palliative treatment. The results showed that patients who received definitive CRT had significantly improved overall survival (50.5% at 5 years) compared to those who received palliative treatment (7.5% at 5 years). The study suggests that more aggressive treatment regimens in select patients may improve survival in oligometastatic EC.
Besides only 36 patients (therefore no multivariate analyses) in single institution and retrospective design, the authors did not mention RT difference over the years.
There have been several advances in radiotherapy technology between 2009 and 2020.
1.IGRT technology allows for the precise delivery of radiation by using imaging to guide treatment. This technology was available in 2009, but there have been significant improvements in the quality and accuracy of imaging since then.
2. IMRT technology was already available in 2009, but its use has become more widespread since then. In addition, VMAT and SBRT. This technology was already available in 2009, but it has become more widely used for certain cancers in recent years.
Overall, these technological advances in radiotherapy have led to more precise and effective treatment with fewer side effects, making it a preferred treatment option for many cancer patients.
Round 2
Reviewer 1 Report
no further comments.
Reviewer 2 Report
I appreciate the authors' prompt reply and have no further questions.